# Long COVID Symptomatology and Associated Factors in Primary Care Patients: The EPICOVID-AP21 Study

**DOI:** 10.3390/healthcare11020218

**Published:** 2023-01-11

**Authors:** Esperanza Romero-Rodríguez, Luis Ángel Perula-de-Torres, Jesús González-Lama, Rafael Ángel Castro-Jiménez, Celia Jiménez-García, Carmen Priego-Pérez, Rodrigo Vélez-Santamaría, Lucía Simón-Vicente, Josefa González-Santos, Jerónimo J. González-Bernal

**Affiliations:** 1Maimonides Institute for Biomedical Research of Córdoba (IMIBIC), Reina Sofía University Hospital, Córdoba University, 14004 Córdoba, Spain; 2Multiprofessional Family and Community Care Teaching Unit of Córdoba, 14011 Córdoba, Spain; 3Multiprofessional Teaching Unit for Family and Community Care, Córdoba and Guadalquivir Health District, 14011 Córdoba, Spain; 4Carlos Castilla del Pino Clinical Management Unit, 14011 Córdoba, Spain; 5“Matrona Antonia Mesa Fernández” Health Center, Cabra Clinical Management Unit, AGS South of Córdoba, 14940 Córdoba, Spain; 6Especialista en Medicina Familiary Comunitaria, University Hospital Reina Sofía, 14004 Córdoba, Spain; 7Faculty of Medicine, University of Córdoba, 14004 Córdoba, Spain; 8Department of Health Sciences, University of Burgos, 09001 Burgos, Spain

**Keywords:** long COVID, persistent COVID, post COVID-19, symptoms, risk factor, admission, vaccination, age, gender

## Abstract

Persistent COVID-19 condition includes a wide variety of symptoms and health problems of indeterminate duration. The present study examined the sociodemographic and clinical characteristics of the population with Long COVID seen in Primary Care using a questionnaire based on the existing scientific literature. It was an observational and descriptive study of the characteristics of the Spanish population with Long COVID over 14 years of age. The responses were analysed by means of a descriptive analysis of the variables recorded, in addition to a bivariate analysis to determine the existence of a relationship between persistent COVID-19 and variables such as gender, age, vaccination status or concomitant pathology. The results obtained clearly describe the sociodemographic characteristics of the population, highlighting the predominance of female gender and the prevalence of tiredness and fatigue. Furthermore, relevant information was obtained on the differences in symptomatology according to gender, age, previous pathologies and alterations derived from infection and/or vaccination. These data are important for better detection, diagnosis and treatment of Long COVID and the improvement of the quality of life of this population.

## 1. Introduction

The COVID-19 pandemic has led to a global health, social and economic crisis. Cooperation between countries belonging to the European Union (EU), as well as the promotion of vaccination against COVID-19 infection, is one of the EU strategies to fight the infection [1].

In Spain, a total of 3,462,593 confirmed cases of COVID-19 have been registered through 14 October 2022 [2]. The clinical characteristics of COVID-19 disease have been addressed in several Spanish [3,4] and international studies [5,6]. The most common symptoms are fever, dry cough, tiredness or fatigue, dyspnoea, sore throat, chills and diarrhoea. Symptomatology described in the acute phase of the disease may persist over time and lead to Long COVID, also called Persistent COVID or Post-COVID syndromes [7].

The World Health Organization defined the post COVID-19 condition as occurring in individuals with a history of probable or confirmed SARS-CoV-2 infection, usually three months after onset, with symptoms lasting at least two months that cannot be substantiated by another diagnosis [8].

The Long COVID condition includes a wide range of chronic health problems of indeterminate duration that can last up to a year, as recent long-term follow-up studies have demonstrated [9]. The prevalence of Long COVID ranges from 10% to 36% in internationals series [10,11,12,13,14]. A recent meta-analysis indicates that the overall prevalence of Long COVID is 43%, with 54% in hospitalised patients and 34% in non-hospitalised patients [15]. However, when considering populations with severe and very severe acute illness and hospitalised patients, the prevalences rise up to 80%

The symptomatology and risk factors have been described in several studies [16,17,18,19,20], as well as by leading institutions [7,21]. The most frequently described symptoms are fatigue, malaise post-exercise, fever, cough, dyspnoea, thoracic pain, palpitations, difficulty concentrating and memorizing, headache, insomnia, dizziness, olfactory and taste disorders, depression or anxiety, diarrhoea, stomach pain, muscular and joint pain and menstrual disorders. The main markers and risk factors for Long COVID appear to be female gender, obesity, recent infection, comorbidities, increased severity at diagnosis and hospitalisation.

Although Long COVID occurs more frequently in people who have been seriously ill, those who have had a mild illness or no symptoms may also be affected [7]. In this respect, a German study [22] noted that Long COVID can occur even after the acute phase in patients with very mild symptomatology who were treated on an outpatient basis. The available scientific evidence is mostly focused on hospitalised cohorts [9,17,20,23], despite the fact that more than 80% of the reported cases of COVID-19 are diagnosed and treated in Primary Care (PC) [24,25].

Therefore, the present study aimed to describe the sociodemographic and clinical characteristics of patients with persistent COVID-19 in the Spanish population treated in PC.

## 2. Materials and Methods

### 2.1. Study Design

An observational, descriptive, case series study of individuals with Long COVID was conducted in the Spanish National Health System, where Long COVID was defined as “a condition occurring in individuals with a history of probable or confirmed SARS-CoV-2 infection, usually 3 months from the onset, with symptoms that last for at least 2 months and cannot be explained by an alternative diagnosis” [8].

### 2.2. Study Population

The study included patients from the general population seen in PC. Patient inclusion criteria were: (a) residence in Spain; (b) age 14 years or older; (c) laboratory diagnosis of acute COVID-19 infection; (d) fulfill Long COVID criteria [8]; (e) consent to participate in the research study.

As for the sample size, based on previous studies estimating the prevalence of Long COVID in the adult population, assuming a percentage of 10% [26], for a confidence level of 95% and a precision of ±3 percentage units, it was estimated that it would be necessary to recruit a sample of at least 385 subjects (calculations made with the Granmo programme: https://www.imim.es/ofertadeserveis/software-public/granmo accessed on 31 May 2021).

The research project has the authorisation of the management of the Cordoba and Guadalquivir Health District and the Cordoba South Health Management Area, as well as the approval of the Clinical Research Ethics Committee of the Reina Sofia Hospital in Cordoba (reference: 5033). Informed consent was requested as part of the online questionnaire. Data processing complied with the provisions of the European Data Protection Regulation and Organic Law 3/2018 on Personal Data Protection and guarantee of digital rights.

### 2.3. Assessments

The information of the participating patients included in the study came from a questionnaire created online with the Google Form tool, from Drive, which was sent to the members of the associations of patients with persistent COVID in Spain through their representatives.

Data on sociodemographic variables (age, gender, employment status and residential area—urban >20,000 inhabitants or rural ≤20,000 inhabitants) were collected. With regard to the clinical features and based on what has been previously published in the scientific literature, a list of 56 possible clinical signs and symptoms for the situation of Long COVID was drawn up, with the signs and symptoms grouped into the following categories: general or non-specific symptoms, body aches, respiratory, digestive, neurological, psychological, ocular, dermatological, cardiovascular and other symptoms. Participants were also asked about their chronic diseases, the treatment they were following for Long COVID, their vaccination status and side effects after vaccination, as well as about hospital and intensive care unit admissions (ICU).

### 2.4. Statistical Analysis

The questionnaire data were completed online in Google Drive by the participants. These data were then exported to an Excel spreadsheet and analysed with the statistical software SPSS version 17.0 (IBM-Inc., Chicago, IL, USA).

First, a descriptive analysis was carried out by calculating absolute frequencies and percentages for qualitative or categorical variables, as well as measures of central tendency, dispersion and position for quantitative variables. Confidence intervals of 95% (95% CIs) were estimated for the main parameters. To calculate 95% CI for qualitative variables, an online calculator (https://www.easycalculation.com/es/statistics/population-confidence-interval.php accessed on 31 May 2021) was used, as SPSS does not have this function. After this, a bivariate analysis was performed to determine whether there was any relationship between the clinical picture of Long COVID, concomitant pathology, hospital admissions, treatment followed or vaccination status, according to the age or sex of the patients.

Age was grouped into 3 categories (qualitative variable: 14 to 40 years, 41 to 52 years and 53 to 76 years), taking into account the limits of the distribution and the percentiles of the dataset. The Chi-square test was used, and differences were considered statistically significant at a value of *p* ≤ 0.05.

## 3. Results

A total of 689 subjects with Long COVID were recruited, of whom 59.5% were diagnosed by PCR test, 30.8% by antigen test and 19.7% by serological test. In 50% of the cases, the family doctor confirmed the diagnosis of Long COVID; in 19.9%, an internist; in 7.5%, a specialist in pneumology; and in the rest of the cases, various specialists, such as a neurologist (3.2%), specialist in Preventive Medicine (1.7%), emergency doctor (1.7%) or cardiologist (1.1%), among others. The mean age of the patients was 45.76 ± 9.67 -SD- (95% CI mean age: 46.03–46.48; range: 14–76 years), and 79.5% were female. A total of 79.3% resided in an urban area (>20,000 inhabitants), 40.9% were on sick leave and 26.9% were not on sick leave at the time, but had been on sick leave as a result of Long COVID.

Exactly 12.2% of the recruited patients reported less than 159 days with Long COVID, 19.1% between 160 and 350 days, 19.2% between 350 and 400 days, 38.0% between 401 and 650 days and 11.5% more than 650 days after being diagnosed with Long COVID.

Table 1 shows the list of clinical signs and symptoms that patients reported to have experienced after they were infected with COVID-19, including those that persisted and those that were most disabling for them with regard to performing activities of daily living.

Tiredness or fatigue were the most frequent (96.8%), persistent (89.4%) and disabling (86.2%) symptoms. There were statistically significant differences in the frequency of persistent symptoms according to age group, with the lowest age group (14–40 years) reporting the highest percentage of fever (21.4%; *p* = 0.035), sore throat (41. 0%; *p* = 0.032), thoracic pain (58.9%; *p* = 0.047), hypothermia (39.9%; *p* = 0.004), shivering (35.4%; *p* = 0.018), memory loss (85.3%; *p* = 0.003), mental fog (82.4%; *p* = 0.020), difficulty swallowing (25.2%; *p* = 0.004) and tinnitus (48.5%; *p* = 0.049). No significant differences were observed in the prevalence of persistent symptoms according to gender, except for fever (more frequent in women, 21.4%; 0.035), and thoracic pain (more frequent in men, 58.9%; *p* = 0.047). On the contrary, 29.8% reported having developed pneumonia after SARS-CoV-2 infection, with no significant differences by age or gender.

A total of 23.4% of patients were admitted to hospital as a result of COVID-19 infection, of which 3.6% had to be admitted to an ICU. No differences were observed with respect to hospital or ICU admission by age or gender.

Figure 1 shows the list of initial symptoms that patients reported at the time they were infected with COVID-19. Headache stands out first (16.0%), followed by fever (13.1%) and sore throat (12.5%), while 9.7% presented unusual tiredness or fatigue.

Table 2 shows the body systems, or apparatuses, that the patients felt to be affected as a result of their Long COVID process. The most frequent cases are those related to mental disorder (73.3%) and nervous system impairment (72.0%). Significant differences by age were observed with cardiovascular (57.9%; *p* = 0.010), endocrine (30.30%; *p* = 0.012), dermatological (36.5%; *p* = 0.013) and haematological (23.0%; *p* = 0.023) system involvement, which were more frequent in the younger age group (14–40 years).

Table 3 shows the chronic diseases or pathologies reported by Long COVID patients. The most frequent health problems were anxiety disorders (45.3%), followed by overweight or obesity (34.1%), depression (27.1%) and hyperlipidaemia (26.9%). No significant differences were found according to age (except in the case of autoimmune diseases, where the prevalence was higher in the 14–40 age group, with 18.0%; *p* = 0.028) or gender.

With regard to the treatments followed by patients in the acute phase, Table 4 shows the list of these treatments. There were no significant differences by age or gender, except for the use of anticoagulant drugs, which was higher among women (13.6%) than among men (2.5%; *p* = 0.005).

Moreover, 44.3% reported drinking alcohol (3.8% regularly and 40.5% sometime) and 10.0% being smokers (6.7% daily and 3.3% sporadically), with no significant differences according to age or gender.

Regarding vaccination status, 12.4% had not received any COVID-19 vaccination, while 88.6% said they had been vaccinated, of which 32.7% stated with one dose, 48.7% with two doses and 18.6% with three doses. A total of 43% were vaccinated with the Pfizer vaccine, 12.3% with Moderna, 6.2% with AstraZeneca and 4.3% with Janssen. Only 0.3% received CoronaVac (from Sinovac). A total of 16.8% were given a combination of Pfizer’s vaccine on one occasion and Moderna’s on another. The most frequent adverse effects attributable to vaccination in the opinion of the respondents are shown in Figure 2. As can be seen, the most frequent adverse reactions were pain at the injection site (90.8%), tiredness or fatigue (76.7%), muscle pain (68.3%), headache (61.8%) and general malaise (58.3%). Other under-reported but more serious side effects were coagulation disorders (5%), facial paralysis (2.3%) and thrombosis (1.8%). Dizziness was more frequent in the 14–40 age group (42.6%; *p* = 0.017), with no further statistically significant differences in the adverse effects perceived by patients according to age or gender.

In total, 63.6% of the subjects reported that their clinical condition remained the same as before vaccination, 12.7% reported improvement after vaccination and 20% reported worsening.

Significant differences were found by age (more subjects aged 14–40 years than older reported worsening, 24.9%; *p* = 0.007) and gender, with a higher percentage of males than females reporting worsening after being administered COVID-19 vaccines (26.9% vs. 18.6%, respectively *p* = 0.045).

## 4. Discussion

The aim of this study was to identify and describe the sociodemographic and clinical characteristics of patients with Long COVID. Long COVID was found to be present in 80% of individuals with a confirmed diagnosis of COVID-19 [25].

Our results show the sociodemographic features that characterize patients with Long COVID, including a mean age of 45.7 years and a predominance of female [26]. The participants were mostly diagnosed by a family doctor using a PCR test. Exactly 23% of the sample reported being hospitalised at the time of acute infection, and 29.8% had pneumonia, with no significant differences by age or gender. Moreover, 38% of the participants reported having symptoms of Long COVID between days 401 and 650 after acute infection, and 40.9% were on sick leave.

Regarding the clinical characteristics of the sample population with Long COVID, our results showed that tiredness and fatigue are key symptoms in terms of the frequency of COVID-19 infection, the persistence of the symptom (leading to the presence of Long COVID) and the disability it produces in the person who has it. As in our results, the review by Salari et al. [27] shows that one of the most common effects of COVID-19 was fatigue, which persisted after acute infection treatment and was present even after 100 days [25,28]. These symptoms can be similar to chronic fatigue syndrome, which includes disabling fatigue with a profound impact on the quality of life of patients [25,27,29]. Next, we find that general malaise, headache, chills and fever are the most frequent symptoms experienced at the time of acute COVID-19 infection [30], although only headache and general malaise were persistent and disabling for some time afterwards. However, the results of the study by Rivera-Izquierdo et al. [31] suggest that the persistence of symptoms should be attributed to the need for hospitalisation, which ultimately prolongs long-term symptomatology, rather than to the COVID-19 infection itself.

Similarly, our results show data on the frequency of persistent symptoms in relation to age, i.e., the younger the age, the higher the frequency of symptoms such as fever, sore throat, thoracic pain, hypothermia, shivering, memory loss, mental fog, difficulty swallowing and/or tinnitus. However, no significant differences were obtained with respect to gender, except for the symptom of fever in women and thoracic pain in men. Currently, most studies do not provide disaggregated data showing significant gender differences, although Sylvester et al. [32] found that women were generally more likely to develop Long COVID, with female gender being a predictor of chronic fatigue and mood disorder symptoms.

It can also be observed that the most affected systems in patients with Long COVID are the mental and the nervous systems, with no significant differences found with respect to gender. According to the literature, the manifestations of Long COVID can affect any area of the nervous system; however, the most frequent symptoms are brain fog, headache, cognitive impairment and sleep and mood disorders, among others [33,34]. These neurological symptoms, or deficits, have been shown to be the main causes of disability in the aftermath of COVID-19 infection [35]. Additionally, according to our results, psychological and mental health problems, such as anxiety or depression, are highlighted as long-term consequences of COVID-19 [27,36,37]. However, taking age into account, there are significant differences in terms of a higher frequency of cardiovascular, endocrine, dermatological and haematological involvement the younger patients were.

There was also a significant prevalence of people with pre-acute infection anxiety disorders, as well as overweight or obesity, depression and hyperlipidaemia. No significant differences were found with respect to gender and age, except for autoimmune diseases, where there was a higher prevalence in the 14–40 age group. Other studies have revealed the significant association of previous health conditions with elevated risk of Long COVID [38]. In agreement with our results, Wu et al. [39] found a significant association between Long COVID and obesity; however, they ruled out the existence of any other significant association with other previous health conditions.

No significant differences were found with respect to gender and age in the treatment applied, except for the use of anticoagulants, where a greater use of anticoagulants was observed in women. Although the use of anticoagulants to improve the sequelae of COVID-19 is still under study, it has been shown to reduce morbidity and improve prognosis [40].

Finally, with regard to vaccination, we can see that the majority of the participating population had been vaccinated with at least a couple of doses, most of them Pfizer, and the most frequent adverse effects were pain at the place of vaccination (90.8%), followed by tiredness or fatigue, muscle pain, headache or general malaise; there were no differences by gender or age. Prevalent but important in terms of severity, adverse effects of coagulation disorders were reported in 5%, facial paralysis in 2.3% and thrombosis in 1.8%. It is also worth noting the presence of significant differences in gender and age with regard to worsening after the application of the vaccine in 24.9% of the subjects aged between 14 and 40 years, mainly males. Currently, although vaccines have been shown to be protective (especially in those vaccinated with two doses) by reducing rates of severe disease or death from COVID-19 [41], they are not as effective in completely preventing disease and prolonged COVID-19 [42]. In addition, several symptoms of prolonged COVID-19 that have overlapped over time with vaccine side effects are being studied. These symptoms include changes in blood pressure, muscle weakness, headaches and brain fog [43,44].

Limitations of this study include the type of survey used, which may introduce a selection bias, as well as an information bias, as it is a self-administered survey. In addition, although symptoms derived from Long COVID are under continuous study, the evidence on the clinical characteristics of Long COVID according to gender and age is scarce, which has been a limitation when contrasting the results obtained.

## 5. Conclusions

In conclusion, our results have described both the sociodemographic and the clinical characteristics of the sample. The predominance of the female gender in Long COVID [45] and the high frequency and prevalence of tiredness and fatigue symptoms in the population after COVID-19 infection were noted [36].

In addition, important information has been obtained about the differences in symptomatology according to gender and age, which is of value for the detection, diagnosis and individualised treatment of patients with Long COVID, according to their characteristics.

Furthermore, it is important to point out and take into account patients’ previous diseases and those that develop as a result of the infection, such as psychological and nervous disorders, underlining the need to complement prevention and/or treatment with multidisciplinary support to improve patients’ well-being, mental health and quality of life.

Although more research is needed, from this study it has been possible to observe some side effects that worsen the health status of patients diagnosed with Long COVID after vaccination, especially in males.

## Figures and Tables

**Figure 1 healthcare-11-00218-f001:**
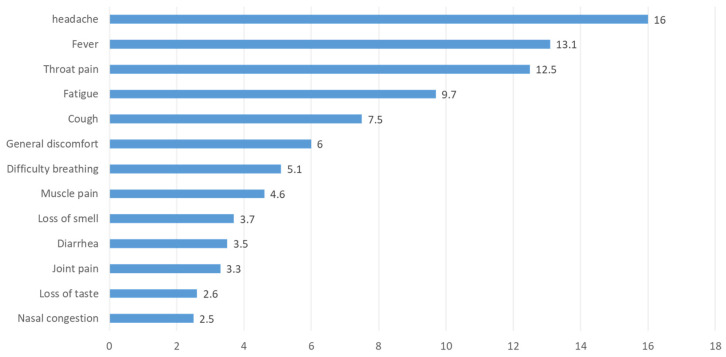
Most frequent first symptoms in the acute phase patients with Long COVID (%).

**Figure 2 healthcare-11-00218-f002:**
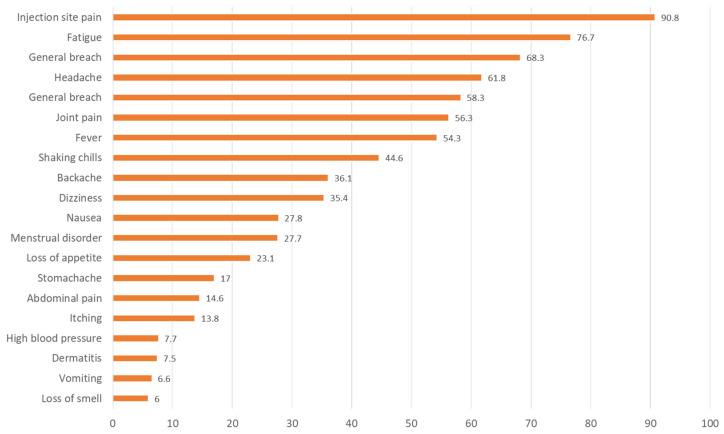
Most frequent adverse events experienced by Long COVID patients after vaccination (%).

**Table 1 healthcare-11-00218-t001:** Symptoms and clinical signs that COVID-19-infected patients presented in their acute phase, those that persisted (Long COVID) and those that were more disabling (*n* = 689).

Symptoms and Signs	COVID-19 *n* %		Long COVID *n* %		Disabling *n* %	
-General and nonspecific symptoms						
General malaise	625	90.7	487	70.7	407	59.1
Tiredness and fatigue	667	96.8	616	89.4	594	86.2
Fever	391	56.7	119	17.3	92	13.4
Low body temperature (hypothermia)	237	34.4	179	26.0	59	8.6
Chills and shivering	409	59.4	218	30.9	79	13.4
Sweating	363	52.7	213	30.9	60	8.7
Headache	593	86.1	474	74.6	360	61.3
Loss of appetite	294	42.7	108	15.7	25	4.8
Itchiness of the skin (pruritus)	281	40.8	193	28.0	52	7.5
Weight loss	282	40.9	105	15.2	22	3.2
-Body aches						
Sore throat	422	61.2	215	31.2	45	6.5
Joint pain	576	83.6	499	72.4	378	54.9
Muscle pain	623	90.4	528	76.6	426	61.8
Thoracic pain	441	64.0	312	45.3	208	30.2
Back pain	478	69.4	372	54.0	243	35.3
Feeling of tightness in the chest	486	70.5	349	50.7	245	35.6
Stomach pain	315	45.7	212	30.8	108	15.7
Abdominal pain	330	47.9	227	32.9	117	17.0
-Respiratory						
Cough	453	65.7	237	34.4	89	12.9
Sputum or phlegm (bronchial secretion)	176	25.5	102	14.8	25	3.6
Hemoptysis (coughing of blood)	31	4.5	11	1.6	5	0.7
Difficulty breathing or shortness of breath	513	74.5	365	53.0	330	47.9
Breathlessness, dyspnea	558	81.0	453	65.7	378	54.9
Nasal congestion (mucus)	314	45.6	176	31.6	54	11.5
Aphonia or hoarseness	304	44.1	185	26.9	82	11.9
-Digestive						
Nausea	320	46.4	190	27.6	100	14.5
Vomiting	122	17.7	40	5.8	33	4.8
Diarrhea	360	52.2	187	27.1	110	16.0
-Neurological						
Dizziness	472	68.5	367	53.3	12	1.7
Vertigo	302	43.8	217	31.5	184	26.7
Tremor	267	38.8	163	23.7	87	12.6
Paresthesia	338	49.1	267	38.8	149	21.6
Loss of smell	366	53.1	141	20.5	51	7.4
Loss of taste	351	50.9	110	16.0	38	5.5
Seizure	31	4.5	21	3.0	12	1.7
Memory loss	533	77.4	484	70.2	399	57.9
Mental confusion	512	74.3	466	67.6	417	60.5
Brain fog	538	78.1	472	68.5	435	63.1
Lack of concentration/attention deficit	604	87.7	543	78.8	476	69.1
Trouble sleeping (insomnia)	521	75.6	431	62.6	321	46.6
Posttraumatic stress	280	40.6	220	31.9	167	24.2
-Ocular						
Conjunctivitis	139	20.2	89	12.9	40	5.8
Blurred vision, foreign body sensation, eye congestion	421	61.6	332	48.1	150	21.8
Dry eyes	323	46.9	272	39.5	112	16.3
-Dermatological						
Hives or eczema on the skin	267	38.8	169	24.4	39	5.7
Facial erythema	138	20.0	76	11.0	13	1.9
Acrosyndrome	121	17.6	72	11.0	32	4.6
-Cardiovascular						
Palpitation	453	65.7	332	48.2	185	26.9
High blood pressure	192	27.9	145	21.0	65	9.4
Low blood pressure	127	18.4	94	13.6	32	4.6
-Others						
Swelling or inflammation in the fingers	202	29.3	141	20.5	56	8.1
Trouble swallowing	193	28.0	123	17.9	52	7.5
Tinnitus (ringing in the ears)	326	47.3	260	37.7	103	14.9
Hair loss	430	62.4	252	36.6	41	6.0
Menstrual disorder (women)	230	33.4	150	21.8	------- -------	
Erection dysfunction (men)	49	7.1	7	1.0		

**Table 2 healthcare-11-00218-t002:** Areas or systems affected as a result of Long COVID (*n* = 689).

Affected Systems	*n*	%	95% CI
Mental disorder (psychological/emotional)	505	73.3	70.0–76.6
Nervous system	496	72.0	68.6–75.3
Respiratory system	455	66.0	62.5–69.6
Digestive system	336	48.8	45.0–52.5
Endocrine system	155	22.5	19.4–25.6
Locomotor system	390	56.6	52.9–60.3
Cardiovascular system	333	48.3	44.6–52.1
Nephrourological system	109	15.8	13.1–18.5
Hematological system (coagulation)	137	19.9	16.9–22.9
Dermatological system	210	30.5	27.0–33.9
Otorhinolaryngological system	223	32.4	28.9–35.9
Eye system	302	43.8	40.1–47.5

**Table 3 healthcare-11-00218-t003:** Diseases and chronic medical conditions present in patients with Long COVID (*n* = 689).

Diseases and Chronic Medical Conditions	*n*	%	95% IC
Arterial hypertension	128	18.6	15.7–21.5
Diabetes Mellitus	31	4.5	3.0–6.0
Chronic Obstructive Pulmonary Disease (COPD).	25	3.6	2.2–5.0
Bronchial asthma	136	19.7	16.8–22.7
Respiratory insufficiency	130	18.9	15.9–21.8
Hyperlipidemia	185	26.9	23.5–30.2
Overweight or obese	235	34.1	30.6–37.6
Immunosuppression (low defense level, HIV, …)	110	16.0	13.2–18.7
Autoimmune disease (ulcerative colitis…)	89	12.9	10.4–15.4
Cancer	8	1.2	0.4–2.0
Kidney failure	24	3.5	2.1–4.9
Heart failure	39	5.7	3.9–7.4
Heart disease (atrial fibrillation, heart valve disease, myocardial infarctions, angina pectoris, left ventricular hypertrophy, …)	62	9.0	6.9–11.1
Cardiovascular disease (vascular accident, stroke, artery disease, …)	35	5.1	3.4–6.7
Anxiety disorder	312	45.3	41.6–49.0
Depression	187	27.1	23.8–30.5
Mental illness (neurosis, psychosis, …)	33	4.8	3.2–6.4
Neurological disease	174	25.3	22.0–28.5
Endocrine disease (hypothyroidism, …)	124	18.0	15.1–20.9
Liver disease	16	2.3	1.2–3.4

**Table 4 healthcare-11-00218-t004:** Drug treatments for patients with Long COVID in acute phase (*n* = 689).

Drugs	*n*	%	95% IC
Analgesics	410	59.5	55.8–63.2
Anti-inflammatories	321	56.4	42.9–50.3
Anxiolytics	193	36.7	24.7–31.4
Antidepressants	218	28.0	28.2–35.1
Anticoagulants	58	8.4	6.3–10.5
Corticosteroids	119	17.3	14.4–20.1
Calcifediol (vitamin D)	290	42.1	38.4–45.8
Other drugs	349	50.7	46.9–54.4

## Data Availability

Informed consent was obtained from all subjects involved in the study.

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
