# Peer review of "Long COVID Symptomatology and Associated Factors in Primary Care Patients: The EPICOVID-AP21 Study"

_healthcare, 2023, doi:10.3390/healthcare11020218_

Round 1

Reviewer 1 Report

 Long COVID symptomatology and associated factors

It seems to me that there is new information in your study that is interesting.

First of all, the number of the patients were small to describe the social health problem.
I don’t understand the clinical implication as the current version.

We need to be aware of the differences between symptoms and prognosis with respect to COVID-19 types.

Furthermore, we want to know the relationships between symptoms (not by vaccination injection) and the presence of vaccination (duration after vaccination).

Because it is important to reveal the effectiveness and adverse effectiveness of vaccination, and the differences of the response to immune system with regard to the age, gender…

The biggest problem is lacking of the clinical implication. How would we apply?
You must analyze and show the detail of the trigger of mental problem, tired ness and cardiovascular diseases.

Author Response

Mr. Rodrigo Vélez Santamaría

Department of health sciences

University of Burgos, Paseo Comendadores s/n.

Burgos, 09001, Spain

Tel. (+34) 947499108

Email: rvs0014@alu.ubu.es

21-12-2022

Healthcare.  Subject: Submissions Needing Revision

Dear editor.

Thank you very much for inviting us to submit our response to reviewers for our manuscript (healthcare-2061364) entitled: “Long COVID symptomatology and associated factors in primary care patients: the EPICOVID-AP21 study”

We have checked our manuscript according to the Academic Editor, the reviewers’ comments and the Journal requirements. We have also responded to some comments from reviewers point by point).

We would be very grateful if you could consider our manuscript to be published in your journal.

Yours sincerely,

Rodrigo Vélez Santamaría, OT, PT

  1. Response to Reviewer 1:

First of all, we would like to express our sincere gratitude for all comments and suggestions received from the Reviewer 1. This information has certainly enriched the text for its best understanding, thank you very much indeed. We have clarified the reviewer1’s questions. We have introduced the required changes both in our answers to the specific comments and in the final manuscript V2.

It seems to me that there is new information in your study that is interesting.

First of all, the number of the patients were small to describe the social health problem.

Response: Thank you for the suggestion. The achieved sample size (n=689) was much larger than the calculated one (n=385), and with that size, and for an expected proportion of 10%, the precision, and therefore the width of the confidence intervals for the 95% confidence of the study parameters would be +/-2.2%, instead of +/- 3.0%. We consider, therefore, that the sample is not small and that it consisted of more than enough patients to describe the social problems with good accuracy.

I don’t understand the clinical implication as the current version.

We need to be aware of the differences between symptoms and prognosis with respect to COVID-19 types.

Response: The results of our study allow us to make progress in the still scarce knowledge that exists on Long Covid-19, of the clinical and epidemiological characteristics of a condition that until very recently was barely recognized, and of which there is still a lack of information on how it manifests itself in different subpopulations, including those captured in the primary care setting, as most of the published studies have been carried out at the hospital or specialized level, and are therefore not very representative of the way this infection is expressed at the community level, with the risk of selection bias, given that those patients who are recruited at the hospital level are the most severe cases (and possibly with a different clinical-epidemiological profile), while the milder cases (which are the vast majority) are less well studied. We believe that descriptive observational studies such as this one have a fundamental role to play when new diseases appear and we do not yet have sufficient knowledge about the presence and frequency of the symptoms and signs that characterize them, and about the type of population that is most affected from a socio-demographic point of view. This will allow us to better orient our actions aimed at early recognition and therapeutic approach.

Furthermore, we want to know the relationships between symptoms (not by vaccination injection) and the presence of vaccination (duration after vaccination).

Response:  Thank you for your comment, bur the duration between symptoms and the presence of vaccination was not recorded in the study and we regret that we are unable to provide the results of such a possible relationship.

Because it is important to reveal the effectiveness and adverse effectiveness of vaccination, and the differences of the response to immune system with regard to the age, gender…

The biggest problem is lacking of the clinical implication. How would we apply?

You must analyze and show the detail of the trigger of mental problem, tired ness and cardiovascular diseases.

Response: Thank you for the suggestion. Although we have already provided above arguments to justify the applicability of the study, they are provided here in more detail:

The present study will have a positive impact on the health of the population, as it will provide a broader and more comprehensive knowledge on the early detection, surveillance and control of COVID-19 disease through the clinical and epidemiological data presented by the general adult population and Primary Care health professionals with COVID-19 disease, with or without persistent clinical symptoms. The clinical and epidemiological characterization proposed in this study is especially relevant in patients with persistent COVID-19, since it is a chronic and disabling pathology at a physical and psychosocial level and requires, like any recently appearing infectious pathology, an exhaustive investigation in the general population and in Primary Care healthcare professionals. The identification of the main characteristics of persistent COVID-19 will allow for better quality care by Primary Care healthcare professionals, a reduction in uncertainty (generated by the lack of knowledge of the disease) in both healthcare professionals and patients, and a better organization and approach to persistent COVID-19 in Primary Care consultations and, ultimately, in the healthcare system.

This research will also provide population-based data on the risk factors associated with persistent COVID-19 disease, which will help to provide more precise information on the magnitude of the risk in PC healthcare staff and in the general population during the advanced phases of the pandemic.

Characterization of the clinical and epidemiological variables of persistent COVID-19 disease in the general adult population and PC healthcare professionals will help to identify which predictors or risk factors are associated with persistent COVID-19, which will help to improve the diagnosis, treatment and follow-up of these population groups. Similarly, the present study will help to prepare and optimize the Health System response in order to achieve better control of the pandemic. Therefore, the findings derived from this work will not only help to understand the direct impact that the disease has had on the health status of the general population and healthcare workers, but will also help to raise awareness among the population and health managers of the crucial role played by primary care healthcare workers in controlling the pandemic. Members of the research team aim to actively disseminate the research project at scientific and social events in order to raise public awareness of the importance of research in this area of knowledge and the role that science and scientists are playing in persistent COVID-19 research. To this end, team members will present the study at the Primary Care Research Open Days, the Science Walk or the European Researchers' Night, which will allow direct communication with the general public and the scientific community.

We hope we have now answered all your comments and we are looking forward to hearing from you again.

Rodrigo Vélez Santamaría, OT, PT

Reviewer 2 Report

Τhis is a very useful study, but it needs significant changes.

-Contemporary literature on the disease reporting high rates of dysautonomia should be taken into account

-neurological and psychiatric data should not be confused

-patients should be separated into hospitalized and non hospitalized (post traumatic stress in hospitalized must be mentioned)

-Conclusions are weak

Please find attached my comments.

Author Response

Mr. Rodrigo Vélez Santamaría

Department of health sciences

University of Burgos, Paseo Comendadores s/n.

Burgos, 09001, Spain

Tel. (+34) 947499108

Email: rvs0014@alu.ubu.es

21-12-2022

Healthcare.  Subject: Submissions Needing Revision

Dear editor.

Thank you very much for inviting us to submit our response to reviewers for our manuscript (healthcare-2061364) entitled: “Long COVID symptomatology and associated factors in primary care patients: the EPICOVID-AP21 study”

We have checked our manuscript according to the Academic Editor, the reviewers’ comments and the Journal requirements. We have also responded to some comments from reviewers point by point).

We would be very grateful if you could consider our manuscript to be published in your journal.

Yours sincerely,

Rodrigo Vélez Santamaría, OT, PT

  1. Response to Reviewer 2:

First of all, we would like to express our sincere gratitude for all comments and suggestions received from the Reviewer 2. This information has certainly enriched the text for its best understanding, thank you very much indeed. We have clarified the reviewer2’s questions. We have introduced the required changes both in our answers to the specific comments and in the final manuscript V2.

Τhis is a very useful study, but it needs significant changes.

-Contemporary literature on the disease reporting high rates of dysautonomia should be taken into account

-neurological and psychiatric data should not be confused

-patients should be separated into hospitalized and non hospitalized (post traumatic stress in hospitalized must be mentioned)

-Conclusions are weak

Response: Thank you for your comments.

We have taken into account all the existing literature to date on this condition. The aim of this study was to give an overall picture of the population with this condition and specifically of each symptom that has been presented when diagnosed with the condition.

The word "psychological" has been eliminated from table 1, leaving only neurological in order to avoid confusion.

In this case, we have not taken into account a specific hospitalised or non-hospitalised cohort; we have taken into account the entire general population, with the participation in the study of all those people with long COVID diangosis, from whom we want to know if they have had symptoms of recent onset after infection with COVID-19.

Our results aimed to describe the clinical and socio-demographic characteristics of the general population diagnosed with long COVID through a sample of people with this condition.

Please find attached my comments.

1.Field 2.1 seems to describe the inclusion criteria rather than the study design. A more detailed description deeded. Which are the questions of survey?

 Response: Thank you for the suggestion. More information has been added in this section.

¨An observational, descriptive and case series study of individuals with Long COVID-19 was conducted in the Spanish National Health System. ¨

  1. Inclusion criteria are missing in 2.2 paragraph

Response: This information has been clarified in the text:

¨Patient inclusion criteria were: a) residence in Spain; b) age 14 years or older; c) laboratory diagnosis of acute COVID-19 infection; d) fulfil Long COVID criteria [8]; e) consent to participate in the research study. ¨ 

  1. By what criterion were the age clusters made?

Response: Thank you for this comment. We used the following criteria to characterize the age of the patients included in the study: the limits of the distribution and the percentiles of the data set.

Attached is the information related to age added to the manuscript:

¨The age was qualitatively determined, creating three categories (from 14 to 40 years, from 41 to 52 years and from 53 to 76 years), considering the limits of the distribution and the percentiles of the dataset¨

  1. Long-COVID is not diagnosed with pcr, rapid test, etc. It obviously means diagnosis of acute infection

Response: Thank you for this thoughtful suggestion. We used the international definition of Long COVID-19, characterized by WHO as “history of probable or confirmed SARS-CoV-2 infection, with symptoms that last for at least 2 months and cannot be explained by an alternative diagnosis”.

This definition is included in the inclusion criteria (Lines 81-84).

  1. It is important to separate the symptoms of the acute phase and the long covid. The recorded symptoms must be presented at least 3 months after the infection. 6. The existence of pneumonia confirms point 5

Response: We appreciate the reviewer suggestion. There are not specific symptoms for the acute phase and the persistent condition of Long Covid. In order to clarified this issue, we showed the different prevalence of the symptoms reported in the acute phase and those reported in the persistent situation of Long COVID-19.

As explained earlier, we used the international definition of Long COVID-19, characterized by WHO as “history of probable or confirmed SARS-CoV-2 infection, with symptoms that last for at least 2 months and cannot be explained by an alternative diagnosis”.

  1. Description of the groups separated in the table 1. Which symptoms considered as disabling? Which symptoms belong to the acute phase and which to Long-COVID?

Response: All symptoms considered belong to long-COVID, i.e. symptoms that have appeared at a specific time and have lasted for more than two months without disappearing. Table 1 shows which of these symptoms have been disabling for people with this condition.

  1. Although the text in line 110 separates the neurological from the psychological symptoms, in this table they are confused. In addition, the phenomenon of dysautonomia, which has been found to be the main problem of long-OVID patients, has not been taken into account. In this panel PTSD is the only phycological/phychiatric and must be in a separate cluster. Also it is reasonable for hospitalized patients to have PTSD while it is strange to find it in asymptomatic or mildly ill patients.

Response: Neurological symptoms have been presented on an individual basis. The phenomenon of dysautonomia has been taken into account, but the aim of this study was to describe the clinical and socio-demographic characteristics of the general population with this condition. Describing the symptoms on an individual basis, in order to find out which symptoms are prevalent, which ones affect the majority of the population and which ones cause the greatest disability.

  1. Figure 1 refers to acute phase symptoms of LC patients?

Response: We appreciate the reviewer comment. Indeed, Figure 1 shows the most frequent first symptoms reported in the acute phase of patients with Long COVID-19 (%)

This information has been clarified in the text:

¨Figure 1 Most frequent first symptoms reported in the acute phase of patients with Long COVID-19 (%). ¨

  1. Which symptoms considered as mental? (Also the word table must be corrected)

Response: Symptoms considered as mental are included in the DSM-V manual, e.g. neurosis, psychosis, bipolar disorder or schizophrenia, among others.

  1. table 3 From the patient numbers as shown in table 3, it appears that almost all patients have some pre-existing disease. Additionally, a large proportion of symptoms may be false positives as they could be present prior to infection due to a pre-existing condition. An example is that a patient with an anxiety disorder will have insomnia, which will be independent of long covid. A possible solution to the problem would be to remove the diseases with symptoms contained in the questionnaire or to do separate analysis-removing patients with anxiety or depression to see the real effect of LC. (see definition of LC in 2.1 paragraph).

Response: Thank you for your comment. This has been taken into account in the evaluation of patients participating in the study. Many of them have previous illnesses, we are aware of this, however, it is important to include them in the study, as many of them have developed long COVID in a more severe, prevalent and disabling way than the population without previous pathology. I understand their confusion regarding symptoms due to the existence of symptoms

derived from previous pathology, however, all participants were diagnosed with long COVID, presenting with new symptomatology (not existing prior to COVID infection) and persisting two months after acute infection. It is also important to note how this condition affects people with a previous pathology and that new symptoms appear after infection and persist over time.

  1. table 4 Is table 4 referred to acute phase treatment?

Response: yes, treatments are applied in the acute phase of the disease. Added in the text

  1. Except vaccination status the vaccination time is also important (before/after LC symptoms)

Response: The questionnaire asks about vaccination status. At the time of the questionnaire all participants are diagnosed with long COVID.

“They were also asked about their chronic diseases, the treatment they were following for Long COVID, their vaccination status and side effects after vaccination, and hospital and intensive care unit admissions”

“Regarding vaccination status, 12.4% had not received any COVID-19 vaccination, while 88.6% reported being vaccinated, of which 32.7% with one dose, 48.7% with two doses, and 18.6% with three doses”

  1. Mental or psychological is not a system. From tables maybe you mean cognitive disorders rather than mental or phycological sphere

Response: Thank you for your comment. The word has been replaced by mental disorder

  1. Almost half of patients have been diagnosed with pre-existing anxiety. From this it cannot be considered that these symptoms are LC symptomatology. The above should be clarified. It is not a novel finding, on the contrary it gives a bias to the study.

Response: Thank you for your comment. This has been taken into account in the evaluation of patients. As mentioned above, it is important to conduct research including all types of population, general population with previous history and pathologies. We have to clarify that all the symptoms taken into account have been new onset symptoms after COVID-19 infection or have been aggravated by COVID-19. We have not taken into account the symptoms previously presented by these patients.

  1. Rather risky and weak conclusion. The study has not been done in the general population but in patients with LC, so it is wrong to generalize this finding for vaccines.

Response: Thank you for your comment and we are sorry we expressed our conclusion poorly. We meant to express the existence of side effects in the general population with this condition.

General revisions This study seems to have been designed without taking into account recent data for LC. Most symptoms reported by the patients are likely to refer to dysautonomia, a disease that has not been considered, though long covid literature shows that the majority of patients suffer from it.

Response: Thank you very much for your comment. It is true according to recent literature that the majority of patients with long COVID have dysautonomia. These data have been taken into account, they are within our knowledge. However, this study has been carried out on a list of 56 possible clinical signs and symptoms collected in the literature, and our aim for this study is to describe in a more concrete and individual way each of these symptoms present in the general population affected and diagnosed with long COVID.

We hope we have now answered all your comments and we are looking forward to hearing from you again.

Rodrigo Vélez Santamaría, OT, PT

Reviewer 3 Report

First i would like to compliment the authors for the time and effort invested in the protocol and manuscript writing.

Overall, the manuscript theory and epidemiological model is well designed. Nonetheless, there are major review points that need to be addressed.

1.     The manuscript needs major language and grammar check. Revision by a native speaker is suggested.

2.     The manuscript is not well cited, major revision is needed. The use of a reference manage is suggested.

Author Response

Mr. Rodrigo Vélez Santamaría

Department of health sciences

University of Burgos, Paseo Comendadores s/n.

Burgos, 09001, Spain

Tel. (+34) 947499108

Email: rvs0014@alu.ubu.es

21-12-2022

Healthcare.  Subject: Submissions Needing Revision

Dear editor.

Thank you very much for inviting us to submit our response to reviewers for our manuscript (healthcare-2061364) entitled: “Long COVID symptomatology and associated factors in primary care patients: the EPICOVID-AP21 study”

We have checked our manuscript according to the Academic Editor, the reviewers’ comments and the Journal requirements. We have also responded to some comments from reviewers point by point).

We would be very grateful if you could consider our manuscript to be published in your journal.

Yours sincerely,

Rodrigo Vélez Santamaría, OT, PT

  1. Response to Reviewer 3:

First of all, we would like to express our sincere gratitude for all comments and suggestions received from the Reviewer 2. This information has certainly enriched the text for its best understanding, thank you very much indeed. We have clarified the reviewer2’s questions. We have introduced the required changes both in our answers to the specific comments and in the final manuscript V2.

First i would like to compliment the authors for the time and effort invested in the protocol and manuscript writing.

Overall, the manuscript theory and epidemiological model is well designed. Nonetheless, there are major review points that need to be addressed.

  1. The manuscript needs major language and grammar check. Revision by a native speaker is suggested.
  2. The manuscript is not well cited, major revision is needed. The use of a reference manage is suggested.

 Response: Thank you for your comments. Improvements have been made to the manuscript.

We hope we have now answered all your comments and we are looking forward to hearing from you again.

Rodrigo Vélez Santamaría, OT, PT

Round 2

Reviewer 3 Report

First I would like to compliment the authors for the time and effort put into addressing the reviewers comments. Overall, the quality of the manuscript improved drastically. Nonetheless, there are still some minor grammar mistakes that need to be addressed before its publication.

For example,

1.      Line 51, “symptoms lasting at least two months that cannot be substantiated another diagnosis” change to , “symptoms lasting at least two months that cannot be substantiated by another diagnosis”

2.      Line 57-59, “However, when considering populations with severe and very severe acute illness and hospitalised patients, then clearly higher prevalences of up to 80% of those affected appear” change to: ““However, when considering populations with severe and very severe acute illness and hospitalised patients, the prevalences rises up to 80%”

Minor grammar mistakes like the previously mentioned, are common throughout the manuscript. Some, lack connectors that make the manuscript hard to understand and some others are not congruent with the paragraph, as some paragraphs are not as detailed.

Therefore, I still recommend a proper traduction and grammar check, reviewed by a certified scientific translator or a native speaker with medical and scientific background. In addition, the link to the questionnaire the authors provide is not active, therefore I recommend to omit the link, as it does not work. However, the questionnaire can be added as supplementary material; I highly recommend adding a table describing the chi-square test value of the variables of the study.

Author Response

  1. Response to Reviewer 3:

First of all, we would like to express our sincere gratitude for all comments and suggestions received from the Reviewer 2. This information has certainly enriched the text for its best understanding, thank you very much indeed. We have clarified the reviewer2’s questions. We have introduced the required changes both in our answers to the specific comments and in the final manuscript V2.

First I would like to compliment the authors for the time and effort put into addressing the reviewers comments. Overall, the quality of the manuscript improved drastically. Nonetheless, there are still some minor grammar mistakes that need to be addressed before its publication.

For example,

  1. Line 51, “symptoms lasting at least two months that cannot be substantiated another diagnosis” change to , “symptoms lasting at least two months that cannot be substantiated by another diagnosis”

Response: Thank you for your comment. Changes have been made.

  1. Line 57-59, “However, when considering populations with severe and very severe acute illness and hospitalised patients, then clearly higher prevalences of up to 80% of those affected appear” change to: ““However, when considering populations with severe and very severe acute illness and hospitalised patients, the prevalences rises up to 80%”

Response: Changes have been made

Minor grammar mistakes like the previously mentioned, are common throughout the manuscript. Some, lack connectors that make the manuscript hard to understand and some others are not congruent with the paragraph, as some paragraphs are not as detailed.

Response: Thank you for your comment. Changes and modifications have been made to the grammar of the whole manuscript.

Therefore, I still recommend a proper traduction and grammar check, reviewed by a certified scientific translator or a native speaker with medical and scientific background. In addition, the link to the questionnaire the authors provide is not active, therefore I recommend to omit the link, as it does not work. However, the questionnaire can be added

as supplementary material; I highly recommend adding a table describing the chi-square test value of the variables of the study.

 Response: Thank you for your comments. Link has been removed. Also, regarding the chi-square you mention, we do not know which variables you are referring to, as it was not included in the previous revision. If you need to add it, please let us know exactly what you are requesting and we will try to give you an answer.

We hope we have now answered all your comments and we are looking forward to hearing from you again.

Rodrigo Vélez Santamaría, OT, PT
